# Hyperoxaluria Induces Endothelial Dysfunction in Preglomerular Arteries: Involvement of Oxidative Stress

**DOI:** 10.3390/cells11152306

**Published:** 2022-07-27

**Authors:** Javier Saenz-Medina, Mercedes Muñoz, Claudia Rodriguez, Cristina Contreras, Ana Sánchez, María José Coronado, Elvira Ramil, Martin Santos, Joaquín Carballido, Dolores Prieto

**Affiliations:** 1Department of Urology, Puerta de Hierro-Majadahonda University Hospital, 28222 Majadahonda, Spain; joaquinalberto.carballido@salud.madrid.org; 2Department of Medical Specialties and Public Health, King Juan Carlos University, 28933 Madrid, Spain; 3Department of Physiology, Pharmacy Faculty, Complutense University, 28040 Madrid, Spain; mmpicos@ucm.es (M.M.); claudrod@ucm.es (C.R.); criscont@ucm.es (C.C.); aasanche@ucm.es (A.S.); 4Confocal Microscopy Facility, Puerta de Hierro-Majadahonda Research Institute, 28222 Majadahonda, Spain; mjcoronado@idiphim.org; 5Molecular Biology and DNA Sequencing Facility, Puerta de Hierro-Majadahonda Research Institute, 28222 Majadahonda, Spain; eramil@idiphim.org; 6Medical and Surgical Research Facility, Puerta de Hierro-Majadahonda Research Institute, 28222 Majadahonda, Spain; martin.santos@salud.madrid.org

**Keywords:** urolithiasis, endothelial dysfunction, oxidative stress

## Abstract

Urolithiasis is a worldwide problem and a risk factor for kidney injury. Oxidative stress-associated renal endothelial dysfunction secondary to urolithiasis could be a key pathogenic factor, similar to obesity and diabetes-related nephropathy. The aim of the present study was to characterize urolithiasis-related endothelial dysfunction in a hyperoxaluria rat model of renal lithiasis. Experimental approach: Endothelial dysfunction was assessed in preglomerular arteries isolated from control rats and in which 0.75% ethylene glycol was administered in drinking water. Renal interlobar arteries were mounted in microvascular myographs for functional studies; superoxide generation was measured by chemiluminescence and mRNA and protein expression by RT-PCR and immunofluorescence, respectively. Selective inhibitors were used to study the influence of the different ROS sources, xanthine oxidase, COX-2, Nox1, Nox2 and Nox4. Inflammatory vascular response was also studied by measuring the RNAm expression of NF-κB, MCP-1 and TNFα by RT-PCR. Results: Endothelium-dependent vasodilator responses were impaired in the preglomerular arteries of the hyperoxaluric group along with higher superoxide generation in the renal cortex and vascular inflammation developed by MCP-1 and promoted by NF-κB. The xanthine oxidase inhibitor allopurinol restored the endothelial relaxations and returned superoxide generation to basal values. Nox1 and Nox2 mRNA were up-regulated in arteries from the hyperoxaluric group, and Nox1 and Nox2 selective inhibitors also restored the impaired vasodilator responses and normalized NADPH oxidase-dependent higher superoxide values of renal cortex from the hyperoxaluric group. Conclusions: The current data support that hyperoxaluria induces oxidative stress-mediated endothelial dysfunction and inflammatory response in renal preglomerular arteries which is promoted by the xanthine oxidase, Nox1 and Nox2 pathways.

## 1. Introduction

Nephrolithiasis is a health problem, which nowadays affects 5–9% of the European population and about 12% of the North American people [1]. It has been pointed out as an independent risk factor for chronic kidney disease (CKD). On the other hand, oxidative stress (OS) is considered an important factor for CKD associated with kidney stones disorder and is a common pathogenic factor with other morbidities linked to nephrolithiasis, such as obesity or diabetes [2]. Endothelial dysfunction (ED) is an early pathogenic event in vascular dysfunction consisting of impaired vasodilation, angiogenesis and barrier function, along with an increased expression of proinflammatory and prothrombotic mediators [3]. ED is involved in diabetes-associated microvascular complications such as diabetic nephropathy [4]. Likewise, epidemiological studies have suggested the development of CKD in non-diabetic obese individuals [5], and experimental studies have demonstrated that ED is a hallmark of the vascular complications associated with obesity [4,6,7]. Oxidative stress plays a pivotal role in the pathogenesis of ED by compromising the availability of nitric oxide (NO), which protects the vascular wall from events leading atherosclerosis. Oxidative stress (OS) has been also pointed out as a key factor associated with ED in diabetic and obesity-related nephropathy and CKD [4,6].

Urolithiasis is an independent risk factor for the development of CKD [8], and obesity and metabolic syndrome contribute to the impairment of renal structure and function that leads to kidney failure when concurrent with lithiasic disorders. Experimental studies have further demonstrated the implication of OS in the exacerbation of the inflammatory response and kidney failure when concurrent with conditions such as obesity and hyperoxaluria [9,10].

A clinical correlation between nephrolithiasis and cardiovascular disorders has recently been reported. Lithiasic patients develop systemic ED as an associated clinical feature [11,12]; furthermore, epidemiological associations occur between nephrolithiasis and cardiovascular morbidities with an odds ratio (OR) between 1.20 and 1.24 [13,14,15]. Moreover, significant relationships have been found between composition of lithiasic patients’ urine and ED, and ED markers have also been found in cell cultures and in animal models of hyperoxaluria [16,17].

In the present study, we aimed to investigate whether OS and inflammation may induce renal ED and thus contribute to the vascular damage in the urolithiasis-associated nephropathy. We used preglomerular arteries from a hyperoxaluria rat model of urolithiasis. The purpose was to clarify the pathogenic changes in the development of nephrolithiasis-associated kidney injury related to OS and ED.

## 2. Materials and Methods

### 2.1. Animal Model

Two groups of 10, 28-day-old Wistar rats were housed in standard conditions, receiving usual chow and water ad libitum for 8 weeks. In the last three weeks, 0.75% ethylene glycol was administered as drinking water to the hyperoxaluric group, as an established model of hyperoxaluria and kidney stone formation [18]. Rats were sacrificed at 12 weeks by slow CO_2_ release in a watertight cage.

Hyperoxaluria model has recently been characterized by our group in previous studies [9,10]. This model develops kidney failure with a significantly lower creatinine clearance compared to the control group, higher percentage of tubules affected by crystal deposits, and interstitial inflammation of the animals affected. Levels of oxaluria were also higher in the OX group [9,10].

All procedures were approved by the institutional animal care committee of Puerta de Hierro Hospital Health Research Institute and the Environment Counseling Department of Madrid province (PROEX 217/17) and conformed the European Guidelines of animal care.

Therefore, two groups were established for comparative analysis: control and hyperoxaluric (OX) (EG 0.75%). The kidneys were quickly extracted and stored in cold physiological saline solutions or in 4 °C RNA later solution.

### 2.2. Dissecting and Mounting of Microvessels

The renal interlobar arteries from control and hyperoxaluric rats were dissected out of the kidney by removing the adjacent medullary connective tissue. Arterial segments were mounted in microvascular myographs (DMT, Hinnerup, Denmark) and equilibrated in PSS maintained at 37 °C. The relationship between passive wall tension and internal circumference was determined for each individual artery. Arteries were set to an internal diameter *L_1_* = 0.9 **L_100_, L_100_* being the internal circumference corresponding to a transmural pressure of 100 mm Hg for a relaxed vessel in situ.

### 2.3. Experimental Procedures for the Functional Assays

At the start of each experiment, arteries were challenged with 120 mM K^+^ (KPSS) to assess vessel viability. The relaxant effects of the endothelial agonist acetylcholine (ACh) were evaluated by the cumulative addition of this agent on arteries precontracted with phenylephrine (Phe) (0.1–0.5 mM). The responses to ACh were then obtained in the absence and presence of the superoxide (SOD) mimetic (tempol, 30 µM), the selective inhibitor of cyclooxygenase 2 (COX-2) NS-398,1 µM, the xanthine oxidase inhibitor (allopurinol, 10 µM), the selective Nox1 inhibitor (NoxA1ds-tat, 0.1 μM) and the selective Nox2 inhibitor (GSK2795039, 1 μM). Arteries were incubated with these drugs before a second concentration–response curve was performed. Vasoconstrictor responses to the α-1 adrenergic agonist Phe were evaluated by cumulative addition of this agent.

### 2.4. Measurement of Superoxide Production by Chemiluminescence

Basal and NADPH-stimulated levels of superoxide (O_2_.^−^) levels were determined by lucigenin-enhanced chemiluminescence in renal arteries and cortex, as earlier reported (Muñoz et al., 2015, 2018, 2020) [6,19]. Renal tissues were dissected from the kidneys of control rat and hyperoxaluric rats, equilibrated in PSS for 30 min at room temperature and then incubated in the absence and presence of the selective COX-2 inhibitor (NS398, 1 µM), the xanthine oxidase inhibitor (allopurinol, 100 µM), the selective Nox1 inhibitor (NoxA1ds-tat, 0.3 μM) or the selective Nox2 inhibitor (GSK2795039, 1 μM) for 30 min at 37 °C, and then stimulated with 100 μM NADPH 15 min prior to determination. Samples were then transferred to microtiter plate wells containing 5 µM bis-N-methylacridinium nitrate (lucigenin). Chemiluminescence was measured in a luminometer (BMG Fluostar Optima), baseline values were subtracted from the counting values for calculation, and superoxide levels were normalized to dry tissue weight.

### 2.5. Quantitative Rt-PCR

Samples of kidney cortex were embedded in RNA during 24 h and were then frozen at −80 °C. RNeasy kit (QIAGEN) Total RNA purification was performed with samples previously homogenized in Trizol solution by the MagNA Lyser System (Roche). The preparations obtained were quantify and quality tested by spectrophotometry. A total of 500 ng of each sample was reverse transcripted to cDNA by using the “First-Strand cDNA Synthesis protocol (NZYtech)”. Real Time Ready Single Assays (Roche) for *Ratus norvegicus* target genes *Nox1* (Assay ID: 506243), *Nox2* (Assay ID: 503271), *Nox4* (Assay ID: 500694), *NF-**κB* (Assay ID: 500911), *Ccl2* (coding for rat MCP-1, Assay ID: 500760), *TNFα* (Assay ID: 502875), and reference gene GAPDH (Assay ID: 503799) were carried out in a LC480 quantitative PCR system under conditions previously described [9,10].

### 2.6. Immunofluorescence Staining

Immunofluorescence analyses were performed to localize and quantify changes in Nox1, Nox2 and Nox4. Kidney samples were fixed in 10% phosphate-buffered formalin in 0.1 M sodium phosphate-buffer (PB), cryoprotected in 30% sucrose in PB, after specimens were embedded in OCT, frozen in liquid nitrogen and stored at −80 °C. Nox1, Nox2 and Nox4 expression were determined by immunofluorescence. Slides were incubated with polyclonal primary antibodies: anti-Nox1, (ab55831), anti-Nox2 (PA-572816), antiNox4 (PA5-114489), from Abcam, Invitrogen and Thermoscientific, respectively, o/n at 4 °C. Then samples were incubated with Alexa Fluor 488 (Invitrogen) secondary antibody. After that, nuclei were stained with To-PRO (T-3605, Thermo-Fisher, Waltham, MA, USA). Slides were mounted with PBS/Glycerol.

Negative controls were processed identically but not primary antibody was used, Images of the specimens were collected with a TCS SP5 confocal microscope (Leica Microsystems, Wetzlar, Germany). The two channels were acquired sequentially to prevent crosstalk between them. The excitation and emission parameters used were (488 nm, 500–540 nm) for Nox1, Nox2 and Nox4 signal and (633 nm, 645–750 nm) for nuclei staining. The gains were adjusted for each channel to prevent saturation in pixels intensity.

### 2.7. Data Presentation and Statistical Analysis

The results of the functional assays are presented as percentage of Phe contraction, as means ± SEM of 4–5 animals. Measurements of superoxide production are expressed as counts per minute (cpm) per mg of tissue of O_2_^.−^ in arterial segments and cortex samples as means ± SEM of 5–12 animals. Results of Rt-PCR for the measurement of expression of the different genes was determined with the delta CT method, calculating the fold difference in expression in relation to the reference gen (GADPH). Statistical differences between means were calculated by Student’s *t*-test or by one-way ANOVA when comparisons involved more than two groups. Statistical differences were established with a level of significance of 95%. The analysis and presentation of the results was performed with a standard software package (Prysm 9.0 Graphpad, San Diego, CA, USA).

## 3. Results

### 3.1. Endothelial Dysfunction in Renal Preglomerular Arteries Is Associated to Augmented Oxidative Stress in Renal Cortex of Hyperoxaluric Rats

Endothelium-dependent vasodilator elicited by ACh were significantly impaired in interlobar arteries from hyperoxaluric rats (Figure 1A, Table 1). Basal superoxide production was enhanced in the renal cortex but not in the arteries from hyperoxaluric compared to control rats (Figure 1B). However, impaired endothelial relaxations in rats from the OX group were improved by acute treatment with the SOD mimetic tempol (Figure 1C,D, Table 2). Vasoconstrictor responses elicited by stimulation of a_1_-adrenoceptors with Phe, or depolarization of VSM with a high K^+^ solution (KPSS) were not significantly different in renal arteries from control and hyperoxaluric rats (Figure 1E, Table 1). The normalized internal lumen diameters, l_1_, were significantly larger in preglomerular arteries from hyperoxaluric compared with control rats, indicating arterial remodeling in the kidney of the OX group.

### 3.2. Role of COX-2 and Xanthin Oxidase in Endothelial Dysfunction of Preglomerular Arteries of Hyperoxaluric Rats

In order to evaluate the mechanisms underlying renal endothelial dysfunction in the kidney of hyperoxaluric rats, the effects of the selective inhibitors of COX-2 and xanthin oxidase, NS398 and allopurinol, respectively, on the relaxations to ACh were assessed. NS398 enhanced ACh relaxations in the renal preglomerular arteries of control rats (Figure 2A, Table 2), thus unmasking the COX-2-dependent contractions induced by the highest concentrations of ACh in these arteries [20]. Treatment with NS-398 also improved endothelial relaxations in arteries from lithiasic rats, although relaxations were not restored to control levels (Figure 2B, Table 2). To further assess the involvement of COX-2 in renal oxidative stress, the effect of the COX-2 inhibitor NS398 on basal ROS generation was examined. NS398 reduced augmented O_2_.^−^ levels in cortical tissue of lithiasic rats (Figure 2C), thus suggesting an enhanced COX-2-mediated oxidative stress in renal cortex of hyperoxaluric rats, while no effects of NS398 were observed in preglomerular arteries from the same animals (Figure 2C). No changes were found either in the levels of COX-2 mRNA expression in arteries from hyperoxaluric compared to control rats (Figure 2D).

In order to assess whether xanthine oxidase is involved in the endothelial dysfunction observed in renal preglomerular arteries of hyperoxaluric rats, the effect of the xanthine oxidase inhibitor allopurinol was examined on the relaxant responses to ACh. Treatment with allopurinol improved endothelial relaxations in renal preglomerular arteries from lithiasic rats and restored them to control levels (Figure 3B, Table 2), but had no effect in control rats (Figure 3A, Table 2). Increased basal O_2_.^−^ levels assessed by lucigenin chemiluminescence in cortical tissue from lithiasic rats were markedly reduced by allopurinol (Figure 3C). However, ROS production in the presence of allopurinol in renal arteries from both groups was similar (Figure 3C).

### 3.3. Role of Nox1, Nox2 and Nox4 in Endothelial Dysfunction and ROS Generation of Renal Preglomerular Arteries from Hyperoxaluric Rats

Involvement of Nox1 and Nox2-derived oxidative stress in the impaired endothelium-dependent relaxations of renal preglomerular arteries in the lithiasic kidney was further investigated by assessing Nox1 and Nox2 expression in arteries of the OX group (Figure 4A,B and Figure 5A,B), and then from a functional point of view, by investigating the effect of the selective inhibitors NoxA1ds-tat and GKT2795039, respectively, on the Ach-induced relaxant responses and ROS generation. Nox1 and Nox2 subunits were significantly augmented in arteries of OX group throughout the arterial wall compared to control arteries (Figure 4A,B and Figure 5A,B). Both Nox1 and Nox2 inhibitors significantly enhanced the relaxations induced by ACh in renal preglomerular arteries of hyperoxaluric rats and restored them to control levels, without significantly altering responses in control rats (Figure 4C,D and Figure 5C,D and Table 2). NoxA1ds-tat and GKT2795039 also inhibited NADPH-stimulated O_2_.^−^ production in renal arteries (Figure 4E and Figure 5E) and cortex (Figure 4F and Figure 5F) of control and OX groups. GKT2795039 exhibited a marked inhibitory effect on the augmented NADPH-dependent oxidative stress in renal cortex from the OX group compared to control rats (Figure 5F). These findings indicate that Nox1 and Nox2-derived ROS are involved in the endothelial dysfunction of renal preglomerular arteries of hyperoxaluric rats.

Expression of Nox4 was further investigated. No differences were found in either Nox4 mRNA or Nox4 protein expression between groups (Figure 6), thus ruling out Nox4 involvement in hyperoxaluria-mediated ED.

### 3.4. Vascular Inflammatory Response in Preglomerular Arteries from Hyperoxaluric Rats (NFκB1, MCP1, TNFα)

In order to assess the inflammatory response elicited by hyperoxaluria in preglomerular arteries, RT-PCR was performed to assess the expression of *NFκB1, MCP1, and TNF**α.*

Hyperoxaluric group showed a higher expression of NFκB1 and MCP-1, although no differences in TNFα expression were found between groups, indicating that hyperoxaluria induces a significant inflammatory response in vascular tissue along with ED and OS (Figure 7).

## 4. Discussion

In this paper, the mechanisms linking urolithiasis and renal ED are first demonstrated by using an established experimental model of nephrolithiasis [21]. Our findings demonstrate that hyperoxaluria induces ED of renal preglomerular arteries in which oxidative stress plays a pivotal role. ED and oxidative stress have been involved in the renal vascular complications of metabolic disorders leading to CKD such as diabetic nephropathy or obesity-associated kidney injury [6,7,20,22,23].

Systemic endothelial dysfunction has been identified in nephrolithiasic patients trough clinical studies, in which the Celermajer method of vasomotor function has been used in brachial artery [11,12,24]. However, ED has not been reported before in the renal vasculature of lithiasic kidneys as a potential risk factor for renal damage either in human studies or in experimental models of renal lithiasis. In the present study, an impairment of the endothelium-dependent relaxations of preglomerular arteries from hyperoxaluric rats has been demonstrated ex vivo by functional studies with microvascular myographs. Furthermore, higher superoxide generation in kidney cortex of the hyperoxaluria group has been identified along with improvement of the relaxation when a general antioxidant tempol was used. These findings thus demonstrate that OS is involved in the development of ED secondary to hyperoxaluria and explain the underlying mechanism linking nephrolithiasis pathology with renal microangiopathy.

Renal vascular dysfunction has also been reported in obesity [20] and diabetes [4], and oxidative stress is now accepted as a key pathogenic factor in the development of ED associated to these metabolic disorders. NO produced by eNOS is rapidly inactivated by reaction with superoxide anions (O_2_^−^) to form peroxynitrite anion (ONOO^−^) [25]. In our study, treatment with the SOD mimetic tempol improved the impaired endothelium-dependent relaxations elicited by hyperoxaluria in preglomerular arteries, thus demonstrating that ROS are involved in hyperoxaluria-mediated endothelial dysfunction. In vitro and animal studies have indicated that diabetes-mediated endothelial dysfunction causes hyperpermeability and plasma leakage at the glomerular level, triggering microalbuminuria which is considered an early stage of kidney damage in diabetes [26,27]. Accordingly, previous studies of our group have demonstrated an impairment of renal function in a rat model of hyperoxaluria that is aggravated by obesity and metabolic syndrome and oxidative stress plays a key pathogenic role in renal inflammatory response and kidney injury [9,10].

Several sources of oxidative stress such as COX, xanthine oxidase, NADPH oxidases and cytochrome P450 (CYP) enzymes have been identified in the nephropathy associated with metabolic diseases [6,28]. The present study demonstrates that impaired vasodilator responses to ACh in preglomerular arteries were associated with higher levels of O_2_.^−^ in the renal cortex of the hyperoxaluric rats, despite both basal and NADPH-dependent O_2_.^−^ production being initially unchanged in preglomerular arteries from lithiasic rats, probably due to a compensatory up-regulation of vascular antioxidant defenses against the hyperoxaluria-induced injury [9,10]. Xanthine oxidase and COX-2 were found to be sources of ROS generation, since both allopurinol and the COX-2 inhibitor NS-398 reverted the enhanced superoxide generation in kidney cortex of the OX group to basal values, although xanthine oxidase inhibition was found to be more effective in restoring the impaired endothelial relaxations of preglomerular arteries from the hyperoxaluric group.

The xanthine oxidase pathway has been involved in vascular oxidative stress and endothelial dysfunction induced by other pathologies such as obesity [20,29], hypercholesterolemia [30] or diabetes [31]. In a rat model of diet-induced obesity, NO-mediation of endothelium-dependent dilation was found to be reduced because of enhanced xanthine oxidase-derived superoxide production [29]. Augmented vascular xanthine oxidase activity in animal models of hypercholesterolemia [30] and improved of ACh-induced dilations in hypercholesterolemic patients treated with oxypurinol [32] have also been reported. It has been proposed that a binding union between a circulating form of xanthine oxidase and endothelial cells could be responsible for enhanced ROS production [33]. In the kidney, allopurinol has been shown to protect glomerular endothelial cells from high glucose-induced ROS generation, indicating that xanthine oxidase is the major source of ROS in diabetic-induced endothelial dysfunction [31]. In preglomerular arteries from genetically obese rats, NO-mediated endothelial dysfunction was also associated to enhanced xanthine oxidase-derived O_2_.^−^ production, contributing to oxidative stress [20]. Interestingly, allopurinol has extensively been used in the preventive treatment of urolithiasis and lithiasic recurrence, and the beneficial effects of this drug could be ascribed to its antioxidant action in the kidney, as shown in the present study.

COX-2-derived oxidative stress has been involved in ED and vascular inflammation in arteries from ageing [34], diabetic [35] and hypertensive rats [36]. Both COX-1 and COX-2 are constitutively expressed in the kidney [37], and their metabolites participate in the physiological regulation of renal blood flow. Nevertheless, COX-2 inhibitors have been shown to improve renal blood flow and to ameliorate hyperfiltration, proteinuria and inflammation in rat models of diabetes [38]. Furthermore, in kidney preglomerular arteries from obese rats, ED is attributable to augmented COX and ROS-mediated endothelium-dependent vasoconstriction and COX-2 inhibition restored impaired endothelial vasodilator responses in arteries from obese rats [20]. In our study, the COX-2 inhibitor NS398 enhanced the ACh-induced relaxant responses of renal arteries in both control and hyperoxaluric rats and ameliorated the higher superoxide generation in kidney cortex of OX group. However, no differences in COX-2 mRNA expression were found between the two groups. On the other hand, enhancement of the endothelium-dependent vasodilator responses by COX-2 inhibition unmasked a COX-2-mediated vasoconstriction in arteries from both the control and OX group [20] but did not restore endothelial relaxations to control levels in the hyperoxaluric group. All together, these results initially rule out a major role of COX-2-derived oxidative stress in urolithiasis-related endothelial dysfunction.

In the present study, enhanced expression of Nox1 and Nox2 was found in preglomerular arteries from hyperoxaluric rats compared to controls, with no changes in Nox4 vascular expression. Furthermore, selective inhibition of Nox1 and Nox2 subunits reduced augmented NADPH-dependent superoxide levels in the renal cortex and improved endothelial relaxations in arteries from the hyperoxaluric group, which suggests that Nox1 and Nox2 are key pathogenic sources of ROS-mediated renal endothelial dysfunction associated to hyperoxaluria. NADPH oxidase is a major source of vascular ROS generation, and Nox2 and Nox1 have been linked to ED in other metabolic and vascular disturbances such as obesity and hypertension [7].

Nox1 subunit is up-regulated in aorta from obese Zucker rats [39], in coronary arterioles from HFD mice [40] and also in kidney preglomerular arteries from obese rats associated with ED [6]. Nox1 activation has been shown to mediate eNOS uncoupling leading to oxidative stress and ED in diabetic mice [41]. On the other hand, we have recently demonstrated augmented Nox1-derived oxidative stress in the renal cortex of hyperoxaluric rats that is aggravated when obesity and hyperoxaluria concur, triggering and exacerbating the inflammatory response in a rat model of combined hyperoxaluria and HFD-induced obesity [10]. Accordingly, the present data show that along with Nox-1-derived oxidative stress in the renal cortex, Nox1 is up-regulated in the vascular wall of preglomerular arteries from lithiasic kidneys and Nox1-derived ROS contribute to ED in these arteries.

Nox2 (gp91phox), the classical pro-inflammatory NADPH oxidase primarily found in phagocytic cells, is up-regulated in the vascular wall and greatly contributes to arterial oxidative stress and ED in metabolic disturbances such as diabetes, obesity and other insulin-resistant states in coronary, cerebral and systemic arteries [42] and also in intrarenal arteries from genetically obese rats [6]. Our results demonstrate that in the hyperoxaluric kidney, Nox2 is also highly expressed throughout the arterial wall both in the endothelium and vascular smooth muscle of kidney preglomerular arteries. Moreover, Nox2-derived ROS from the renal cortex contribute to renal endothelial dysfunction, as depicted from the inhibitory effect of Nox2 blockade in reducing superoxide production and restoring endothelium-dependent relaxations. On the other hand, our group has recently demonstrated that Nox2 and Nox4 are physiologically sources of H_2_O_2_ generation that contribute to the endothelium-dependent vasodilation of human renal arteries; thus, Nox subunits play a protective role in the kidney vasculature [19]. The present study shows that endothelial Nox2 mRNA expression widely spreads throughout the vascular smooth layer, and its contribution to the endothelial relaxations of preglomerular arteries is impaired in hyperoxaluria, thus hindering Nox protective effects on renal endothelium. Conversely, Nox4 expression was unchanged in kidney arteries from hyperoxaluric rats. While some authors have reported an up-regulation of Nox4 expression in hyperoxaluria [43] and that Nox4 has been associated to oxidative stress in diabetic nephropathy and renal injury, our data are consistent with those in earlier studies suggesting no contribution of vascular Nox4 to kidney oxidative stress and showing lower levels of Nox4 coupled to impaired Nox4-derived H_2_O_2_ generation in preglomerular arteries from obese rats [6] and a reduced expression of renal tubule Nox4 and H_2_O_2_ generation in hyperglycemic type 1 diabetic mice and other models of chronic kidney disease [44].

Nox1 and Nox2 subunits, in particular Nox2, are closely related to vascular inflammation and have been involved in oxidative stress in insulin resistant states, obesity and atherosclerosis, wherein a chronic low-grade vascular inflammation is associated with the redox-sensitive NF-κB inflammatory signaling pathway [45]. Endothelial cells can modulate the immune reaction by the recruitment of inflammatory cells through the induction of leucocyte adhesion molecules, cytokines or ROS [7]. ED is associated to an augmented expression of proinflammatory mediators as a result of NF-κB induction. Our study revealed a significant elevation NF-κB and MCP-1 expression, suggesting an inflammatory response induced by ROS in preglomerular arteries from hyperoxaluric rats. NF-κB is comprised of a tightly controlled system regulating the inflammatory and redox states of vascular smooth muscle and endothelial cells [46]. The NF-κB pathway has been implicated in both age- [47] and obesity-associated vascular ED [48]. Additionally, it has been shown that NF-κB plays an important role in mediating vascular ED in overweight and obese middle-aged and older humans [49]. On the other hand, renal epithelial cell exposure to oxalate, a constituent of the most common form of kidney stones, promotes a rapid degradation of IκBα, an endogenous inhibitor of the NF-κB transcription factor [50].

MCP-1 urinary excretion has often been used as an index of renal inflammatory status [51] and has also been related to ED in obesity and hypertension rat models [52]. Oxalate exposure also induces expression of MCP-1 in cultured renal cells and has been suggested to be responsible for the leucocyte influx that leads to tubular inflammation and renal damage [53]. In our study, hyperoxaluria was also found to develop an enhanced expression of MCP-1 in the preglomerular vessel. Thus, hyperoxaluria induces an inflammatory kidney status in both vascular and tubular territories, which can be co-responsible for the development of kidney failure.

The inflammatory response triggered in the obesity-related kidney injury or in the diabetic nephropathy by NF-κB includes a significant increase of proinflammatory markers in renal tissue, such as TNFα and MCP-1 [54,55]. Previous studies performed by our group in a hyperoxaluria model also demonstrate higher TNFα expression in the renal cortex [10]. On the other side, NF-κB has been shown to directly regulate the expression of genes encoding proinflammatory cytokines TNFα, IL-6 and c-reactive protein in endothelial cells [48,49]. In contrast to our previous study performed in the renal cortex, no significant differences have been found in preglomerular arteries TNFα expression between the groups, despite a trend to increase in arteries from hyperoxaluric rats was observed. Nevertheless, our results show an inflammatory response triggered by NF-κB in which MCP-1 expression raises in hyperoxaluric rats.

Systemic ED has been demonstrated in lithiasic patients as an alteration of the endothelial relaxations of peripheral arteries [11]. This feature is considered as a predictor of cardiovascular events and an explanation of the epidemiological relationship achieved between urolithiasis and cardiovascular morbidity [13,14]. In the current study, ED was further demonstrated in preglomerular arteries from hyperoxaluric animals in a rat model of urolithiasis. ED might thus contribute to the renal damage associated with urolithiasis, as it does in diabetic or obesity-associated nephropathy, and may be an important cause of the additive deleterious effect on renal structure and function reported when both pathologies converge triggering significant kidney damage [10].

## 5. Conclusions

In conclusion, the present study demonstrates that hyperoxaluria triggers oxidative stress-mediated impairment of the endothelial relaxations and a vascular inflammatory response in kidney preglomerular arteries, hallmarks of renal endothelial dysfunction. Xanthine oxidase is a source of ROS in the renal cortex that triggers ED in preglomerular arteries. Since xanthine oxidase inhibition restores endothelial vasodilatation, it could be considered as a therapeutic strategy in lithiasis-associated kidney damage. Nox1, and especially Nox2, trigger an oxidative stress response in the renal cortex, which generates endothelial dysfunction in the nearby arteries. Again, the inhibition of both ROS sources restores these alterations. Oxidative stress response includes an inflammatory vascular response through the activation of the NF-κB pathway as a typical feature associated to endothelial dysfunction.

This study first reports renal ED secondary to hyperoxaluria. Likewise, in other metabolic disturbances leading to CKD such as diabetic or obesity-related nephropathy, ED could be in part responsible for kidney injury in lithiasic patients. Prevention and treatment of ED should improve the renal function of these patients.

## Figures and Tables

**Figure 1 cells-11-02306-f001:**
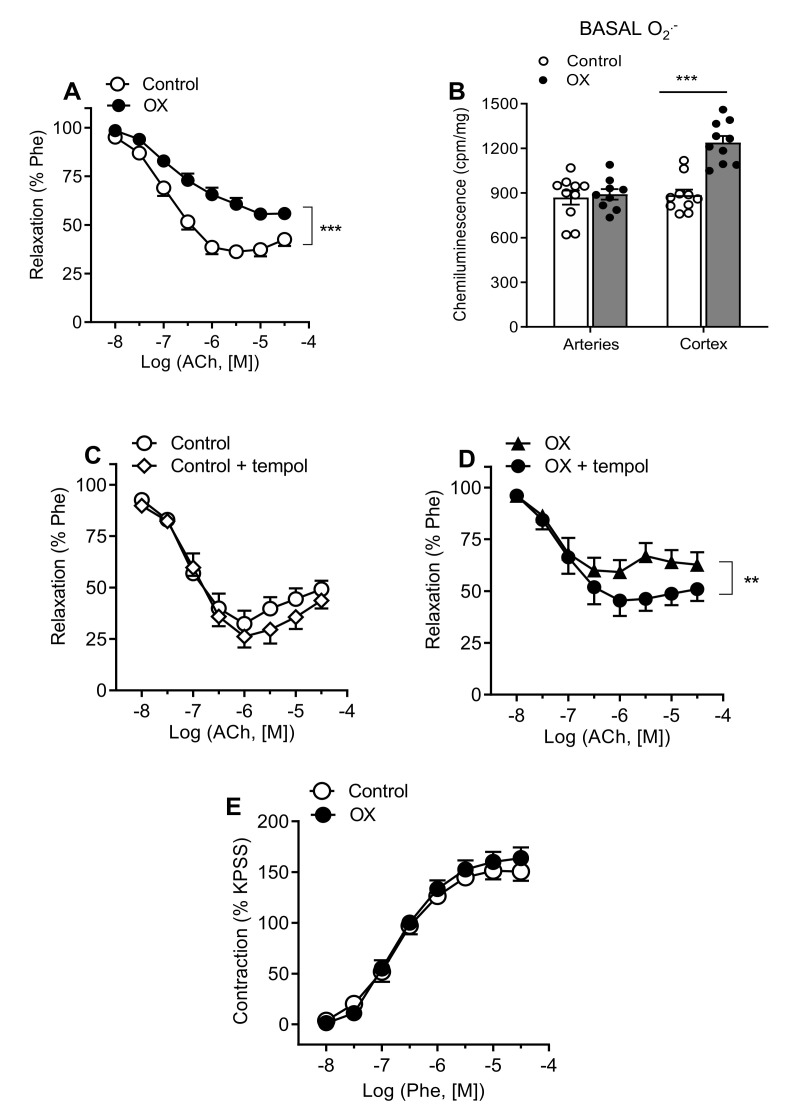
Impaired endothelium-dependent relaxations of the renal preglomerular arteries are associated with higher basal O_2_^−^ generation in the renal cortex in hyperoxaluric rats (OX) and ameliorated by the SOD mimetic tempol. (**A**) Comparative endothelium-dependent relaxations of intrarenal arteries levels in control and OX rats. Results are given as a percentage of the phenylephrine (Phe)-induced contraction. Data are means ± SEM of 4 animals (1–2 arteries from each rat). Statistical differences were calculated with unpaired *t*-test *** *p* < 0.001. (**B**) Basal ROS generation in kidney cortex and renal interlobar arteries measured by lucigenin-enhanced chemiluminescence. Results are given as counts per minute (cpm) per mg of tissue. Bars represent mean ± SEM of 5 animals. Statistical differences were calculated with unpaired Student’s *t*-test *** *p* < 0.001 versus basal levels. (**C**,**D**) Effect of tempol on the relaxant responses elicited by ACh in rat renal interlobar arteries in control (**C**) and OX groups (**D**). Statistical differences were calculated by unpaired Student’s *t*-test ** *p* < 0.01. (**E**) Constrictor responses to Phe in renal interlobar arteries from control and hyperoxaluric rats (OX). Results are expressed as absolute values (Nm^−1^). Data are shown as the mean ± SEM of 7–8 arteries (4 animals).

**Figure 2 cells-11-02306-f002:**
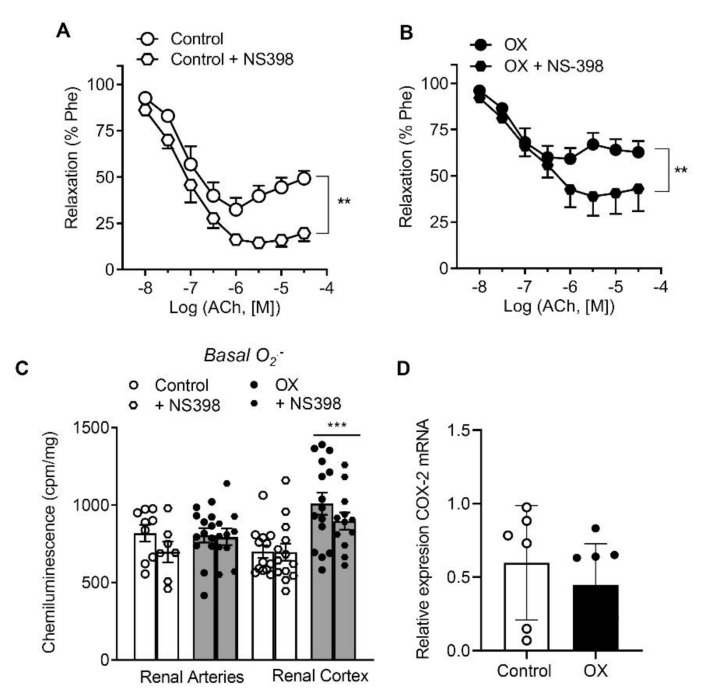
COX-2 inhibition improved renal endothelial relaxations in both control and OX rats and reduced elevated O_2_^−^ levels in the renal cortex of the OX group, with no differences in COX-2 mRNA expression between the two groups. (**A**,**B**) Comparative effects of NS398 on endothelial relaxations of rat intrarenal arteries in the Control (**A**) and OX groups (**B**). Results are given as a percent of the phenylephrine (Phe) contraction as the mean ± SEM of 7–8 arteries (4 animals). Statistical differences were calculated by paired *t*-test ** *p* < 0.01. (**C**) Effect of NS398 on basal ROS generation in the renal arteries and kidney cortex of control and hyperoxaluric rats, measured by lucenin-enhanced chemiluminescence. Results are expressed as counts per minute (cpm) per mg of tissue. Bars represented mean ± SEM of 5–10 animals. Statistical differences were calculated with *t*-test *** *p* < 0.001 versus basal levels. (**D**) Comparative COX-2 mRNA expression in the renal arteries of the control group (*n* = 6) and OX group (*n* = 8), performed by RT-PCR. The delta–delta CT method was used to determine the fold change gene relative expression in the arteries that were treated with water (control group) and with 0.75% EG (OX group). The values represent the mean ±  SD. No statistical differences were detected.

**Figure 3 cells-11-02306-f003:**
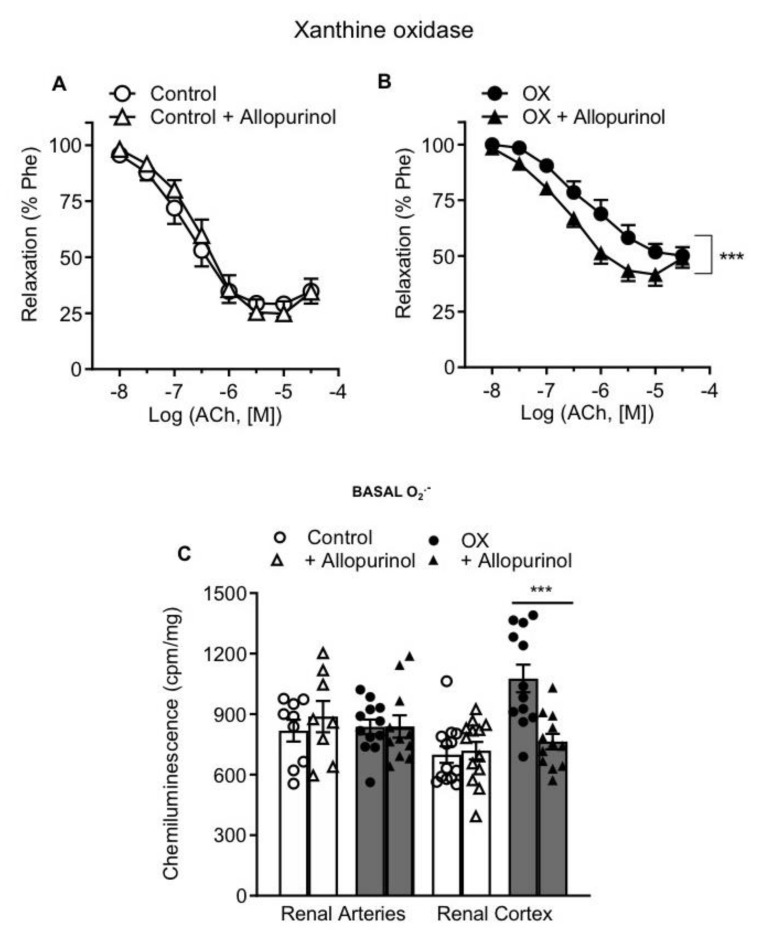
Xanthine oxidase inhibition reduced elevated O_2_^−^ levels in the renal cortex from the OX group and improved impaired endothelial relaxations in the OX group. (**A**,**B**) Effect of the xanthine oxidase inhibitor allopurinol on the endothelial relaxations of rat intrarenal arteries in the control (**A**) and OX groups (**B**). Results are given as percentage of the phenylephrine (Phe) contraction as the mean ± SEM of 4 animals (1–2 arteries per animal). Statistical differences were calculated by paired Student’s *t*-test *** *p* < 0.001. (**C**) Effect of allopurinol on basal ROS generation in the kidney cortex and renal interlobar arteries measured by lucigenin-enhanced chemiluminescence. Results are expressed as counts per minute (cpm) per mg of tissue. Bars represent mean ± SEM of 5–10 animals. Statistical differences were calculated with *t*-test *** *p* < 0.001 versus basal levels.

**Figure 4 cells-11-02306-f004:**
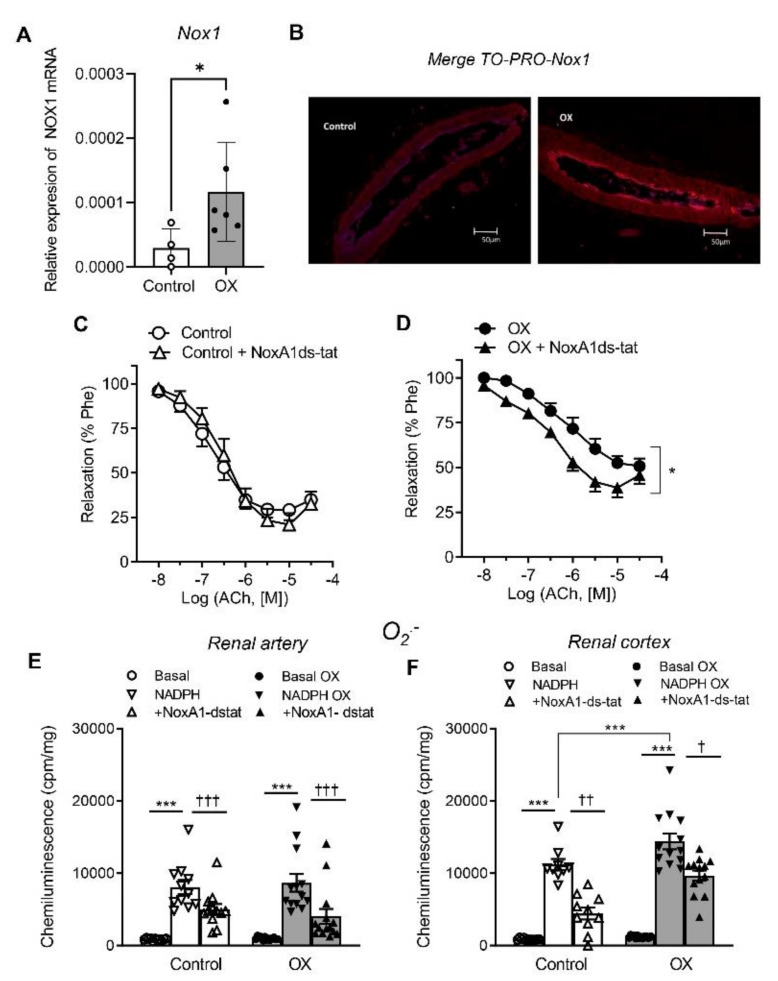
Enhanced Nox1 mRNA expression and amelioration of elevated renal cortex NADPH-dependent O_2_^−^ levels and impaired endothelial relaxations by selective Nox1 inhibition in preglomerular arteries from hyperoxaluric rats. (**A**) Comparative Nox1 mRNA expression in samples of renal arteries from the control (*n* = 4) and OX group (*n* = 6), performed by RT-PCR. The delta CT method was used to determine the fold change for Nox1 gene relative expression in the arteries that were treated with water (control group) and with 0.75% EG (OX group). Statistical differences between means were calculated by unpaired Student’s *t*-test * *p* < 0.05. (**B**) Immunofluorescence demonstration of Nox1 subunit protein in a section of a renal interlobar artery. Immunofluorescence double labeling for TO-PRO marker (blue areas) demonstrates nuclear staining and for Nox1 protein (red areas). (**C**,**D**) Comparative effects of the Nox1 inhibitor NoxA1ds-taT on the endothelial relaxations of rat intrarenal arteries in control (**C**) and OX groups (**D**). Results are given as a percent of the phenylephrine (Phe) contraction as the mean ± SEM of 8 arteries (4 animals). Statistical differences were calculated by an unpaired Student’s *t*-test *** *p* < 0.05 (pEc_50_, see Table 2). (**E**) Effects of NoxA1ds-tat on NADPH-dependent ROS generation in renal interlobar arteries and in kidney cortex from control and hyperoxaluric rats. (**F**) measured by lucigenin chemiluminescence. Results are expressed as counts per minute (cpm) per mg of tissue. Bars represented mean ± SEM of 5–10 animals. Statistical differences were calculated with *t*-test *** *p* < 0.001, versus basal levels. † *p* < 0.05, †† *p* < 0.01; ††† *p* < 0.001 versus NADPH-stimulated.

**Figure 5 cells-11-02306-f005:**
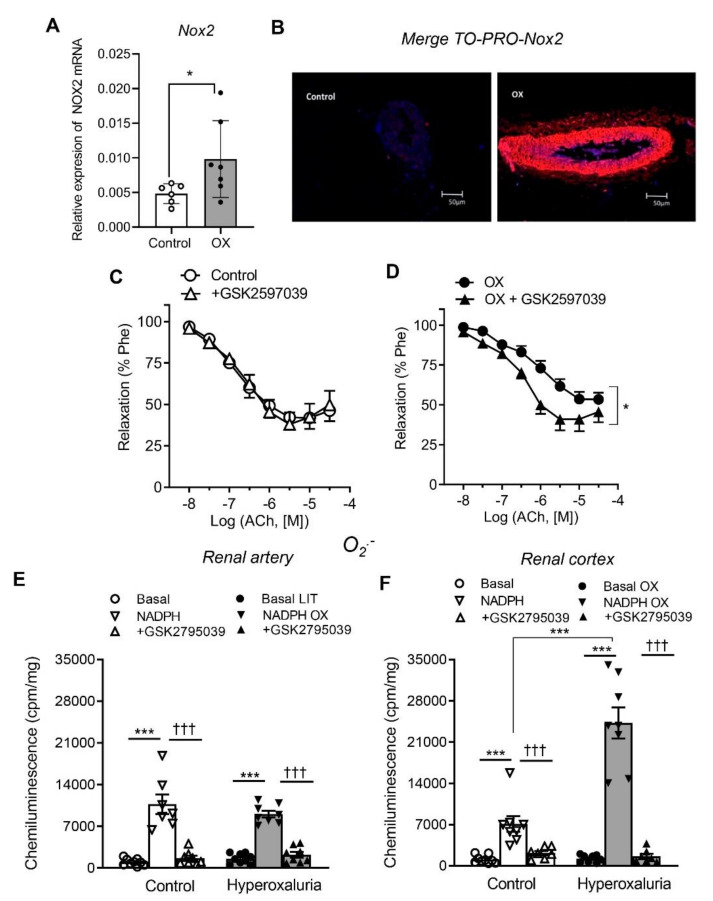
Up-regulation of Nox2 and amelioration of oxidative stress in renal cortex and endothelial dysfunction by selective Nox2 inhibition in preglomerular arteries from hyperoxaluric rats. (**A**) Comparative Nox2 mRNA expression in samples of renal arteries from the control (*n* = 6) and OX group (*n* = 6) performed by RT-PCR. The delta CT method was used to determine the fold change for Nox2 gene relative expression in the arteries that were treated with water (control group) and with 0.75% EG (OX group). * *p* < 0.05. (**B**) Immunofluorescence demonstration of enhanced expression Nox2 protein in a section of a renal interlobar artery from OX group compared to a control artery. Immunofluorescence double labeling for TO-PRO marker (blue areas) demonstrates nuclear staining and for Nox2 protein (red areas). (**C**,**D**) Comparative effects of the Nox2 inhibitor GSK2795039 on endothelial relaxations of rat intrarenal arteries in control (**C**) and OX groups (**D**). Results are expressed as a percent of induced contraction. Data are shown as the mean ± SEM of 4 animals. Statistical differences were calculated by a Student’s *t*-test for paired observations * *p* < 0.05 versus control. (**E**,**F**) Effects of inhibitor GSK2795039 on NADPH-dependent ROS generation in renal arteries and cortex measured by lucigenin-enhanced chemiluminescence. Results are given as counts per minute (cpm) per mg of tissue. Bars represented mean ± SEM of 5–10 animals. Statistical differences were calculated by Student *t*-test *** *p* < 0.001 versus basal levels. ††† *p* < 0.001 versus NADPH-stimulated levels.

**Figure 6 cells-11-02306-f006:**
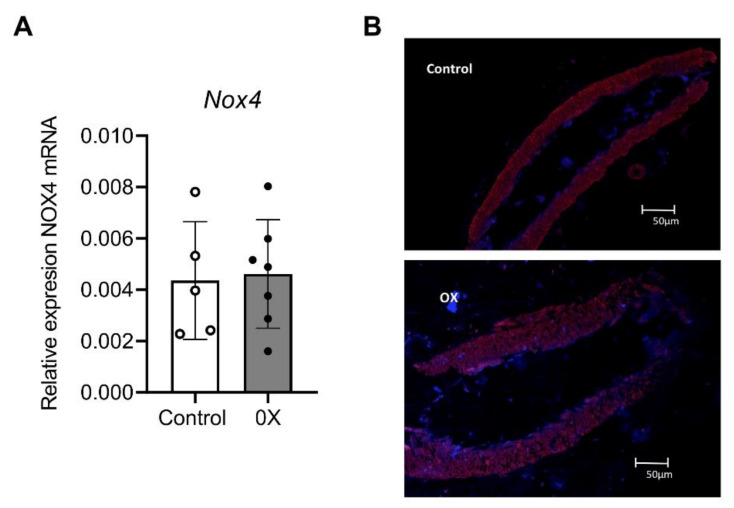
Unchanged Nox4 mRNA levels in preglomerular arteries from hyperoxaluric rats. (**A**) Comparative Nox4 mRNA expression in samples of renal arteries from the control (*n* = 5) and OX group (*n* = 7), performed by RT-PCR. The delta CT method was used to determine the fold change for NOX4 gene relative expression in the arteries that were treated with water (control group) and with 0.75% EG (OX group). Values represent the mean ± SD. Statistical differences were calculated by a Student’s *t*-test for unpaired observations. No significant differences were found. (**B**) Immunofluorescence demonstration of NOX4 protein in a section of a renal interlobar artery from the Control and OX group. Immunofluorescence double labeling for TO-PRO marker (blue areas) demonstrates nuclear staining and for NOX4 protein (red areas).

**Figure 7 cells-11-02306-f007:**
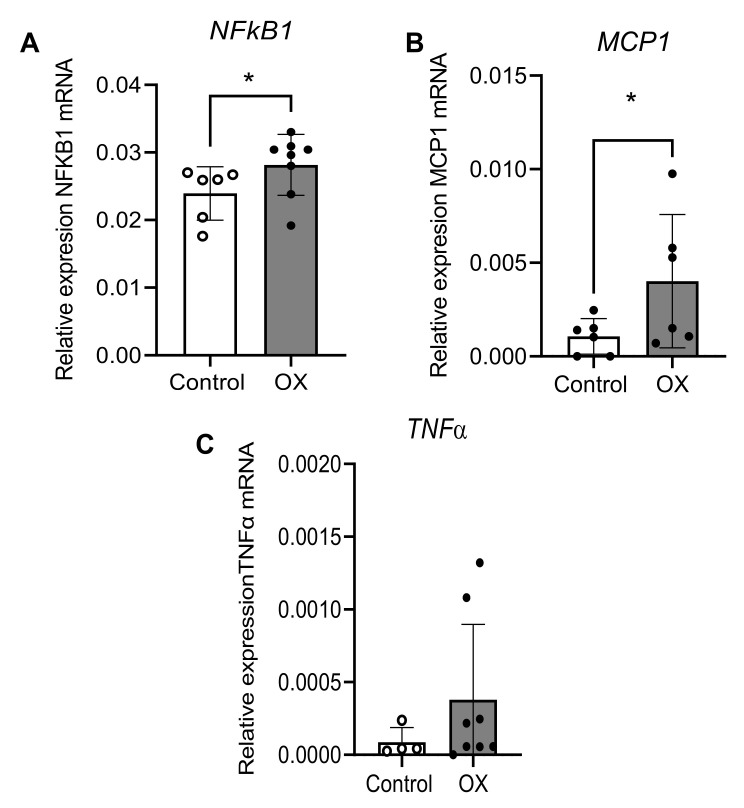
NFKB1 and MCP-1 are up-regulated in renal interlobar arteries of the OX group. Comparative *NFKB1* (A) *MCP-1* (B) and *TNFα* (C) mRNA expression in samples of renal arteries from control (*n* = 5–6) and OX group (*n* = 7–8), performed by RT-PCR. The delta CT method was used to determine the fold change for the different genes’ relative expression in the arteries that were treated with water (control group) and with 0.75% EG (OX group). Values represent the mean ± SD. Statistical differences were calculated by a Student’s *t*-test for unpaired observations * Significantly different at *p*-value < 0.05.

**Table 1 cells-11-02306-t001:** Vasorelaxant and contractile responses of renal arteries from control and hyperoxaluric rats.

	CONTROL		*Relaxation*		OX			
	*pEC_50_*	*Emax (%)*	*l_1_*	*n*	*pEC_50_*	*Emax (%)*	*l_1_*	*n*
Ach	6.84 ± 0.07	69 ± 3	295 ± 13	22	6.25 ± 0.12 ***	50 ± 3 ***	360 ± 14 **	30
			*Contraction*					
	*pEC_50_*	*Emax (Nm^−1^)*	*l_1_*	*n*	*pEC_50_*	*Emax (Nm^−1^)*	*l_1_*	*n*
Phe	6.80 ± 0.13	3.09 ± 0.50	299 ± 14	12	6.73 ± 0.03	3.30 ± 0.38	360 ± 26 *	15
KPSS	-----	2.36 ± 0.33	297 ± 14	14	-----	2.14 ± 0.26	360 ± 26 *	15

Data are means ± S.E.M; *n* is the number of individual arteries, 1–2 per animal. *pEC_50_* is –logEC_50_, EC_50_ being the concentration of agonist giving 50% of the maximum effect (*Emax*). *l_1_* is the normalized lumen diameter in µm. Significant differences between means were analyzed by unpaired Student’s *t*-test. * *p* < 0.05 ** *p* < 0.01 *** *p* < 0.001 vs. control rats.

**Table 2 cells-11-02306-t002:** Effects of the SOD mimetic tempol (30 µM), the COX-2 inhibitor NS-398 (1 µM), the xanthine oxidase inhibitor allopurinol (10 µM), the Nox1 inhibitor NoxA1ds-tat (0.1 µM) and the Nox2 inhibitor GSK2795039 (1 µM), on the concentration-relaxation curves to ACh in renal arteries from control and hyperoxaluric rats.

	CONTROL				OX			
	*pEC_50_*	*Emax (%)*	*l_1_*	*n*	*pEC_50_*	*Emax (%)*	*l_1_*	*n*
Control	7.02 ± 0.10	65 ± 6	286 ± 16	7	6.98 ± 0.11	44 ± 5 ^‡‡^	367 ± 43	9
+tempol	6.96 ± 0.11	71 ± 6		7	7.09 ± 0.13	56 ± 6 **		9
Control	7.02 ± 0.10	65 ± 6	286 ± 16	7	6.92 ± 0.12	45 ± 6 ^‡‡^	376 ± 48	8
+NS-398	7.19 ± 0.21	84 ± 3 **		7	7.07 ± 0.15	62 ± 10 **		8
Control	6.76 ± 0.13	74 ± 5	267 ± 23	9	6,01 ± 0.15 ^‡‡^	55 ± 4 ^‡‡^	354 ± 13	13
+allopurinol	6.58 ± 0.12	78 ± 6		9	6,69 ± 0.18 **	59 ± 5		13
Control	6.69 ± 0.16	74 ± 6	259 ± 24	7	5.99 ± 0.16^‡‡^	53 ± 5 ^‡‡^	354 ± 13	12
+NoxA1ds-tat	6.50 ± 0.13	80 ± 7		7	6.44 ± 0.13 *	62 ± 5		12
Control	6.80 ± 0.09	65 ± 7	345 ± 14	6	6.01 ± 0.21 ^‡‡^	48 ± 4 ^‡^	324 ± 24	7
+GSK2795039	6.82 ± 0.15	66 ± 8		6	6.54 ± 0.10 *	67 ± 8 *		7

Data are means ± S.E.M; *n* is the number of individual arteries of 4–5 animals. *pEC_50_* is –logEC_50_, EC_50_ being the concentration of agonist giving 50% of the maximum effect (*Emax*). *l_1_* is the normalized lumen diameter in µm. Significant differences between means were analyzed by paired or unpaired Student’s *t*-test. * *p* < 0.05, ** *p* < 0.01 vs. control in the absence of treatment; ^‡^ *p* < 0.05; ^‡‡^ *p* < 0.01 vs. control non hyperoxaluric rats.

## Data Availability

The data presented in this study are available in this article.

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
