# Peer review of "Hyperoxaluria Induces Endothelial Dysfunction in Preglomerular Arteries: Involvement of Oxidative Stress"

_cells, 2022, doi:10.3390/cells11152306_

Round 1

Reviewer 1 Report

C

1)    In the text, the words “hyperoxaluric rats” are used. Did they measure blood biochemistry? Renal function? Renal morphology? How was damage in the kidney in particular in vessels (glomeruli) developed? An H&E can show the damage in the kidney. Damage in glomerular arteries may cause decreased GFR.

2)    In fact, it is mitochondria that produces major ROS, although other pathways are also involved e.g. NADPH, COX, Xanthine oxidase. Did they have any data to show damage of mitochondria in arteries? What reason for ROS production? The mechanism by which NADPH (COX/xanthine oxidase) pathway is activated in arteries seems not examined, is it due to uric acid or oxalate?

3)    The quality of figure 4a, c, d, e, f is not as good as figure 1-3, also figure 5-7, please revise them. Also the size of graphs is variable, please make the size of graphs consistent.

4)    Figure 4-6b are shown as immunofluorescence, but not mRNA, please correct relevant description in the text. Figure 3 xanthine, not xanthin

5)    OX should show full name when it appears the first time (page 2). P16 line 371, it is a incomplete sentence.

6)    Please check typos and grammar

Author Response

Attached bellow is a file with comments to reviewer 1

Reviewer 2 Report

Please attach an explanation of the abbreviations.

Please check that the purpose of the work corresponds to the obtained results and conclusions.

Low legibility of the charts Fig4. And Fig 5.

There is no information on the severity of acidosis and calcium levels,

What was the concentration of oxalate in the blood of rats?

What was the EGFR of rats?

Line 381; that hyperglucemia-mediated endothelial dysfunction

Line 456-466: write it more clearly.

Overall Impression - An interesting topic, but the paper is hard to read. I propose to review the text and try to simplify it.

Author Response

Attached bellow is a file with comments to reviewer 2.

Round 2

Reviewer 1 Report

I don't have further comments